# Building a Bridge between Chemotherapy and Immunotherapy in Malignant Pleural Mesothelioma: Investigating the Effect of Chemotherapy on Immune Checkpoint Expression

**DOI:** 10.3390/ijms20174182

**Published:** 2019-08-26

**Authors:** Elly Marcq, Jonas RM Van Audenaerde, Jorrit De Waele, Julie Jacobs, Jinthe Van Loenhout, Glenn Cavents, Patrick Pauwels, Jan P van Meerbeeck, Evelien LJ Smits

**Affiliations:** 1Center for Oncological Research, University of Antwerp, 2000 Antwerp, Belgium; 2Department of Pathology, Antwerp University Hospital, 2650 Antwerp, Belgium; 3Department of Pulmonology & Thoracic Oncology, Antwerp University Hospital, 2650 Antwerp, Belgium; 4Center for Cell Therapy and Regenerative Medicine, Antwerp University Hospital, 2650 Antwerp, Belgium

**Keywords:** chemotherapy, tumor microenvironment, mesothelioma, immune checkpoints, MPM, PD-1, PD-L1, TIM-3, LAG-3

## Abstract

In light of the promising results of immune checkpoint blockade (ICPB) in malignant pleural mesothelioma (MPM), we investigated the effect of different chemotherapeutic agents on the expression of immune checkpoints (ICPs) in order to rationally design a good treatment schedule for their combination with ICP blocking antibodies. Cisplatin, oxaliplatin and pemetrexed are interesting chemotherapeutic agents to combine with immunotherapy given their immunomodulatory capacities. We looked into cisplatin and pemetrexed because their combination is used as first-line treatment of MPM. Additionally, the effect of the immunogenic chemotherapeutic agent, oxaliplatin, was also studied. Three different MPM cell lines were used for representation of both epithelioid and sarcomatoid subtypes. The desired inhibitory concentrations of the chemotherapeutic agents were determined with the SRB-assay. Allogeneic co-cultures of MPM cells with healthy donor peripheral blood mononuclear cells (PBMC) were set up to assess the effect of these chemotherapeutic agents on the expression of ICPs (PD-1, LAG-3, TIM-3) and their ligands (PD-L1, PD-L2, galectin-9). Cisplatin might be a promising treatment to combine with ICP blocking antibodies since our MPM cell lines were most susceptible to this stand-alone treatment. We found that the expression of ICPs and their ligands on both MPM cells and PBMC was mostly downregulated or unaltered when treated with chemotherapeutic agents, though no clear trend could be determined.

## 1. Introduction

Malignant pleural mesothelioma (MPM) is an aggressive cancer that affects the membranes lining the lungs. It is causally associated with occupational asbestos exposure and characterized by a long latency period (20−40 years between the inhalation of asbestos fibers and MPM presentation) [1,2,3]. Over the years, global incidence has increased steadily with the highest annual rates reported in Australia, Great Britain and Belgium [2]. Due to its long latency period and the high manufacturing/ usage of asbestos in the late nineties, incidence rates are expected to increase in the following decades [4] and thus MPM will remain a global health problem. The overall survival and prognosis of MPM patients is very poor with an average survival time of 12 months for untreated patients and a 5-year survival rate of less than 5% [4,5,6,7]. At the moment, the combination of a platinum-compound (cisplatin) and an anti-folate (pemetrexed) is used as first-line treatment for MPM in the clinic. The combination of both compounds led to a significant increase in median overall survival of approximately 3 months and an increase in total response rate of approximately 30% in comparison with cisplatin stand-alone treatment (12.1 months vs 9.3 months and 41.3% vs 16.7%). These responses suggest a synergism between cisplatin and pemetrexed [8]. Although this chemotherapeutic strategy plays a very important role in the treatment of MPM, its impact on median overall survival is limited [9]. The medium survival might be improved through combination with different treatment modalities. For example, the combination of immunotherapy (pembrolizumab) with chemotherapy led to a significant improvement of progression-free survival (from 9 months to a median of 13 months) and higher overall response rates in patients with NSCLC [10,11].

Cancer immunotherapy research entered a new phase with the discovery of immune checkpoints (ICPs), immune modulatory molecules, operating via a series of inhibitory pathways, responsible for maintaining immune homeostasis and modulating immune responses, thereby preventing autoimmunity [12]. Overexpression of ICPs reduces antitumor immune responses and is suspected to result in an immune resistant tumor [12]. Therefore, abolishing tumor-immune resistance by blocking ICPs became a promising approach in the field of cancer immunotherapy.

In this study, we aimed to investigate the effect of chemotherapeutic agents on the ICP expression of immune cells and MPM tumor cells in order to develop a rational treatment schedule for the combination of chemotherapy with ICP blockade (ICPB). When looking into combination strategies, it is of utmost importance to rationally design a good treatment schedule. Pemetrexed and cisplatin are widely used in the treatment of different types of cancer. Due to the suspected synergistic effect of cisplatin in combination with pemetrexed [8,13,14] and the fact that it is used as first-line treatment of MPM patients, we investigated this combination strategy in our study. In addition, oxaliplatin was also included due to its ability to induce immunogenic cell death (ICD) [15]. ICD distinguishes itself from physiological cell death because of the induced cell structure changes (such as the increased expression of tumor-associated antigens or the emission of danger signals that stimulate immune responses), which allow immune cells to detect these cancer cells and initiate an anticancer response [16]. On top of that, oxaliplatin has recently shown to be less toxic and more tolerable compared to other chemotherapeutics in gastric cancers [17].

To date, the combination of chemotherapy with immune checkpoint blockade in MPM has not been described, and therefore, it is not yet known when immunotherapy should be introduced in the treatment scheme. By looking at the effect of different chemotherapeutic agents on the immune checkpoint expression profile of MPM tumor cells and immune cells, we hope to gain more knowledge on which treatment schedule might be best suited for the combination of immunotherapy with chemotherapy. For example, if chemotherapy upregulated the expression of ICPs it might be best to give immunotherapy as an adjuvant treatment. However, in case chemotherapy downregulates ICP expression it would be better to give immunotherapy as a neo-adjuvant treatment. First, we determined the effective concentrations of different chemotherapeutics (cisplatin, oxaliplatin and pemetrexed) to treat mesothelioma cell lines. Then, allogeneic co-cultures of MPM cells with healthy donor peripheral blood mononuclear cells (PBMC) were set up. The effect of these chemotherapeutic agents on the expression of ICPs (PD-1, LAG-3, TIM-3) and their ligands (PD-L1, PD-L2, galectin-9) in this co-culture system were assessed with flow cytometry. Analysis with FlowJo provided more detailed information on the effects of the different treatments.

## 2. Results

### 2.1. Cytotoxicity of Chemotherapeutics Is Cell Line Dependent

In order to investigate the effect of chemotherapy on immune checkpoint expression in MPM we first determined the chemotherapeutic concentrations for the treatment of MPM cells. Different concentrations ranges were evaluated for cisplatin (0−50 μM), oxaliplatin (0−50 μM) and pemetrexed (0−10 μM), and chemosensitivity was assessed with the Sulphorodamine B (SRB) assay. Three cell lines were used to represent the most common subtype of MPM, epitheliod (NCI-H2818, NCI-H2795), and the most lethal subtype, sarcomatoid (NCI-H2731). Statistical analysis showed significant differences in susceptibility of our MPM cell lines to the investigated chemotherapeutic agents (Figure 1). In general NCI-H2818 cells were more sensitive to cisplatin (*p*-values between 0.001 and 0.020) and oxaliplatin (*p*-values between 0.001 and 0.009) compared to NCI-H2795 and NCI-H2731. All cell lines showed no response to pemetrexed.

Calculation for the inhibitory concentration (IC) values were performed for each agent. Table 1 summarizes the IC50 values which clearly reflects the varied sensitivity between the cell lines. NCI-H2818 was significantly more sensitive to cisplatin and oxaliplatin compared to NCI-H2731 (*p* = 0.007, *p* = 0.030, respectively) and NCI-H2795 (*p* = 0.008, *p* = 0.001, respectively). NCI-H2731 was also more sensitive to oxaliplatin compared to NCI-H2795 (*p* = 0.012). As reflected by the lack of response in Figure 1. our MPM cell lines were not sensitive to pemetrexed. However, IC values for pemetrexed were determined previously in our lab on pemetrexed sensitive cancer cell lines [18] and therefore we decided to use those values for further experiments.

The combination of cisplatin and pemetrexed commonly performed as the first-line therapy in the clinic, was also investigated on the MPM cell lines. The effects on cell survival were compared to stand-alone treatments using the SRB analysis. In order to delineate the effects of the future combination strategies with ICPB, we worked with the IC20 values of cisplatin, pemetrexed and oxaliplatin (Table 2). For pemetrexed the IC20 value of 0.05 μM was used, based on existing literature [18].

Our results, shown in Figure 2, confirmed the earlier observed sensitivity of our cell lines to cisplatin. Compared to our untreated controls all cell lines showed approximately 20% decreased cell survival after 72 h of cisplatin treatment and no response to pemetrexed. There were no significant differences in cell survival following treatment compared to the untreated control, nor were there differences within the treatment groups.

### 2.2. Chemotherapeutics Have A Variable Influence On ICP Expression

In order to rationally design a treatment schedule for the combination of chemotherapy with immune checkpoint blockade, we investigated the effect of our different chemotherapeutics on the expression of three immune checkpoints (programmed death-1 (PD-1), lymphocyte activation gene-3 (LAG-3) and T-cell immunoglobuline-3 (TIM-3)) along with their corresponding ligands (programmed death ligantd-1/2 (PD-L1/2) and galectin-9) using multicolor flow cytometry (FCM). The expression on both MPM cells and PBMC were investigated after being in co-culture for 72 h.

The mean percentages of positive cells and the change in mean fluorescence intensity (ΔMFI values) (Figure 3 and Figure 4, respectively) were compared between the treated and the untreated group. Varying results in effect were observed on ICP expression of both MPM cells and PBMC. When comparing the immune checkpoint expression of the treated groups with the untreated group, only significant differences were noted for the TIM-3 expression (% positive cells) on PBMC in co-culture with NCI-H2731 after cisplatin treatment (*p* = 0.037, Figure 3). No other significant differences were found for the percentage of cells expressing immune checkpoints (% positive cells, Figure 3) or for the intensity of immune checkpoint expression (ΔMFI, Figure 4). Based on these results, no solid conclusion can be drawn regarding the best treatment schedule for the combination of chemotherapy and immune checkpoint targeting.

## 3. Discussion

To date, MPM remains a health problem due to its poor prognosis and limited clinical benefit of currently used treatments. Taken together, this highlights the need for novel treatment strategies. Multimodal approaches that combine different treatments (e.g., chemo-immunotherapy, combined ICP blockade) are emerging due to more favorable outcomes compared to single-modality treatments. Benefits of multimodal treatments have already been confirmed within MPM by the results of the phase III EMPHACIS trial indicating potential synergism between cisplatin and pemetrexed. Results of this trial led to the approval of simultaneous administration of cisplatin (75 μg/m^2^) and pemetrexed (500 μg/m^2^) as a first-line treatment for MPM patients since February 2004. This still remains the current standard of care [8]. Although the combination of cisplatin and pemetrexed has shown clinical benefit compared to cisplatin stand-alone treatment, response rates are still limited. In the MAPS1 trial, the anti-angiogenic drug bevacizumab was added to cisplatin + pemetrexed treatment which resulted in improved overall survival compared to the chemotherapy alone, thereby confirming the benefits of multimodal treatments [19]. Since promising results regarding ICPB are recently reported within MPM, including improved overall survival rates and acceptable safety and tolerability profile [20,21,22], this treatment might also offer new opportunities for combination strategies.

In this study, we evaluated the cytotoxic effects of different chemotherapeutic agents on the NCI-H2731, NCI-H2795 and the NCI-H2818 MPM cell lines. All MPM cell lines showed differences in susceptibility to cisplatin and oxaliplatin, which highlights the heterogenous and complex nature of MPM in terms of treatment susceptibility. IC50 values of different chemotherapeutics have already been reported on other MPM cell lines [23,24,25], though we are the first to describe the IC values of cisplatin and oxaliplatin for the three MPM cell lines used in our experiments. A cell line-dependent IC50 value was observed for cisplatin and oxaliplatin, an observation that is consistent with data from other research groups [23,24,25,26]. In MPM, different histological subtypes can be distinguished of which the sarcomatoid subtype is suggested to be more resistant to several chemotherapeutic agents [27]. In addition, a systematic review described less response to chemotherapy in patients with sarcomatoid MPM compared to other histological subtypes [28]. With regard to cisplatin + pemetrexed, only 4 out of 18 patients with sarcomatoid MPM subtype responded to therapy. Combinations with oxaliplatin showed even less response in sarcomatoid MPM patients (1/5 responded to vinorelbine + oxaliplatin and no response was shown to raltitrexed + oxaliplatin) [28]. Our data showed a similar sensitivity to cisplatin for the sarcomatoid NCI-H2731 and the epithelioid NCI-H2795 cell lines, while the latter was the least sensitive to oxaliplatin. The epithelioid NCI-H2818 cell line was the most sensitive for both cisplatin and oxaliplatin. We also noted an intrasubtype variation between our two epithelioid cell lines regarding the IC values for both chemotherapeutics. While the NCI-H2818 already showed response at low concentrations of chemotherapy, higher concentrations were used for the NCI-H2795 to reach the same response. Taken together, our data demonstrate that sensitivity to chemotherapy, more specifically to cisplatin and oxaliplatin, might not only depend on the histological subtype. As reported for survival by Rosen et al. [29] and Pelosi et al. [30], it might be that response to chemotherapeutic treatment also depends on nuclear grading of their tumor.

Regardless of the concentration, we found that the three cell lines showed no response to pemetrexed, suggesting that they are resistant to pemetrexed. So far, nothing has been described about pemetrexed resistance in the NCI-H2731, NCI-H2795 and NCI-H2818 MPM cell lines and its IC values for these 3 cell lines has never been reported. Therefore, the underlying mechanism that explains the lack of in vitro response to pemetrexed remains unclear. A possible explanation might be the overexpression of thymidylate synthase (TS) protein, as reported in different studies for non-small cell lung cancer (NSCLC), colon cancer cell lines as well as MPM [31,32,33,34,35]. Pemetrexed exerts its function through inhibition of DNA synthesis-related enzymes such as TS. In TS-overexpressing cells, it is suggested that pemetrexed fails to exert this function [31,32,34]. However, to confirm this hypothesis, further research is required. On the other hand, only a 41.3% response rate is noted in MPM patients for the first-line treatment with cisplatin and pemetrexed [8], suggesting that more than half of the patients might be chemotherapy resistant. It might be that our MPM cell lines are derived from patients who were pemetrexed-resistant.

When comparing the effect of cisplatin as a stand-alone treatment to its combination with pemetrexed, we did not observe significant differences, although a synergy between both compounds has been suggested [8]. Since a synergy between both compounds has only been reported in in vivo studies and clinical trials, it might be possible that this effect was not observed in our in vitro experiment due to limitations of our model. However, an in vitro study investigating cross-resistance of cisplatin and pemetrexed in three epitheliod MPM cell lines (NCI-H2452, NCI-H28 and NCI-H226) and one biphasic cell line (MSTO-211H) reported a synergistic effect of cisplatin + pemetrexed in three out of four cell lines [36]. An additive effect (i.e., the effect is the sum of cisplatin and pemetrexed treatment) was found for the fourth cell line (NCI-H2452) which also seemed to be the most resistance against both cisplatin and pemetrexed.

Antagonistic as well as synergistic effects of combining cisplatin and pemetrexed have also been reported in several cancer types. Kim et al. [37] found synergism in two out of 6 gastric cancer cell lines and an additive effect for three others. On the other hand, Kano et al. [38] reported a schedule dependent interaction between cisplatin and pemetrexed in different carcinoma cell lines. Simultaneous as well as sequential (cisplatin followed by pemetrexed) administration of the two agents produced antagonistic effects in three out of four carcinoma (lung, breast and ovarium) cell lines. An additive effect was observed in a colon cancer cell line. In contrast, 24 h of pemetrexed treatment followed by 24 h of cisplatin produced additive or synergistic effects in all cell lines [38].

There are some limitations when comparing our results with the ones from the previously mentioned studies. First of all, different protocols and cell lines were used. Secondly, different concentrations of chemotherapeutic agents were used that sometimes even did not lie within the range of the clinically admitted therapeutic doses. Thirdly, to exclude that chemotherapy does not influences the trypsin sensitivity of our ICPs and their ligands it would be ideal to compare results with another methods that is optimized for the detection of our antigens of interest. Finally, there are only few in vitro studies describing a potential synergism between cisplatin and pemetrexed. The synergistic effect has been primarily based on results obtained from clinical trials in MPM. Therefore, further research with regard to the cisplatin/pemetrexed working mechanisms is required.

Since many chemotherapeutic agents have been suspected to be able to exert an immunomodulatory effect on the immune system in addition to their cytotoxic effect [39,40,41,42], there is an increasing interest in combining them with immunotherapy. Immunotherapy can be given as a neoadjuvant, concomitant (simultaneous) or adjuvant treatment to chemotherapy. Gaining more insight in the effect of chemotherapy on the expression of immunotherapeutic targets might help to define a good treatment schedule. In general, overexpression of immune checkpoints -which is observed in several cancer types [12] is suspected to create a defense-line against the antitumor immune response of the host. One can suggest that when the tumor is therapeutically targeted, it appeals to this defense-line in order to improve tumor cell proliferation by upregulating the expression of immune checkpoints which results in immune suppression. However, this hypothesis is not confirmed by our in vitro data. We observed a lot of interexperimental variation as far as the effect of chemotherapy on immune checkpoint expression is concerned, suggesting that the immune checkpoint expression is very dynamic. Moreover, for each experiment a different PBMC donor was used which might explain the difference in ICP expression on immune cells. ICP expression can be induced by both tumor cell intrinsic signals as well as extrinsic signals such as the secretion of IFN-γ. The fact that T-cell activity with corresponding IFN-γ secretion can differ between donors might explain the variability in expression. In general, although statistically not significant, we noticed a trend towards decreased immune checkpoint expression in two out of three MPM cell lines when treated with either cisplatin, oxaliplatin or pemetrexed. However, since this was not the case for our third cell line, no firm conclusions could be drawn and further investigation is warranted.

The mechanisms by which cisplatin, oxaliplatin and pemetrexed can regulate the expression of PD-1 and its ligands PD-L1 and PD-L2 have not yet extensively been described. Even less is known about the effect of these chemotherapeutic agents on other immune checkpoints and their ligands such as LAG-3, TIM-3 and galectin-9 [43]. With regard to the influence on PD-L1 expression, cisplatin has been shown to increase the expression in hepatoma cell lines through MEK-ERK (MAPK) signaling [44]. On the other hand, altered PD-L1 expression is reported to be mainly driven by the activation of Akt and signal transducer and activator of transcription 3 (STAT3) pathways [45]. Other research groups described the involvement of these pathways in chemotherapy-induced increased PD-L1 expression on dendritic cells (DC) and NSCLC cell lines [46,47]. In general, multiple chemotherapeutic agents have been described to influence PD-1 and/or PD-L1 expression. Paclitaxel was reported to induce PD-L1 cell surface expression on breast cancer cells as well as on ovarian cancer cells [48,49]. On top of that, other research groups noticed enhanced expression of PD-L1 and PD-1 on leukemia cells after decitabine treatment [50] and increased PD-L1 expression on breast cancer cells after doxorubicin treatment [51]. Furthermore, other results were reported by Oki et al. [52] who described a decrease of PD-1 expression in normal lymphocytes of patients with Hodgkin lymphoma after panobinostat treatment. Sheng et al. [53] investigated the PD-L1 expression pre and post chemotherapeutic treatment in NSCLC tissue samples on both tumor cells and tumor-infiltrating immune cells. A significant decrease in PD-L1 expression was described in the tumor cells after chemotherapeutic treatment. Interestingly, this was only observed for the patients who responded to treatment. Data previously obtained within our lab also show similar results that confirm our in vitro data. PD-L1 expression was only observed in unpretreated MPM tissue samples which suggest that chemotherapy downregulates PD-L1 expression [54]. Results were obtained using immunohistochemistry.

It should be mentioned that there are some limitations when comparing our own results of cisplatin and oxaliplatin treatment with those reported by the studies mentioned above. Not only were different techniques used (i.e., immunohistochemistry versus FCM), but the effect on expression was examined in other tumor types. Furthermore, data of the currently available studies report observations made on immortalized cell lines as well as patient samples embedded in paraffin, while our data are based on the effect of chemotherapy in an allogeneic co-culture of MPM cells with healthy donor PBMC. The effect of the PBMC on the immune checkpoint expression of MPM cells and vice versa should also be considered. As it stands, the contradictory results reported in literature concerning the effect of platinum-compounds on immune checkpoint expression could be explained by the heterogeneity among malignancies and the therapeutic agents used. Concerning pemetrexed, we are the first to describe its effect on immune checkpoint expression. Reduced immune checkpoint expression was observed in the three MPM cell lines after treatment with pemetrexed. Since high expression levels of PD-L1 are correlated with a reduced sensitivity to cisplatin treatment [47,55], pemetrexed might influence the effect of cisplatin treatment. When combined with cisplatin, pemetrexed might stimulate the effect of cisplatin by reducing the expression of PD-L1, resulting in increased cytotoxicity compared to cisplatin as a stand-alone treatment. However, this hypothesis cannot be confirmed by our cytotoxicity data, which showed no significant difference in cell survival between the two groups. Our FCM data do not support this hypothesis either, since they showed a decrease in PD-L1 expression in two out of three cell lines following pemetrexed treatment.

Taken together, we are the first to describe a small effect of cisplatin, pemetrexed, oxaliplatin and cisplatin + pemetrexed on the immune checkpoints LAG-3 and TIM-3 as well as the TIM-3 ligand, galectin-9 in vitro. Similar to our obtained data, lowered TIM-3 expression was also described in diffuse large B-cell lymphoma after chemotherapeutic treatment. In contrast to our study, reduced TIM-3 expression was noted after sequential administration of different chemotherapeutic agents (rituximab, cyclophosphamide, doxorubicin, vincristine and prednisone) [56]. Since a high interexperimental variability with regard to the effect of chemotherapy on immune checkpoint expression was observed in our study, we cannot draw a firm conclusion about the best treatment schedule for combining immunotherapy with chemotherapy in MPM. Further preclinical testing for this combination strategy is required to identify the most promising regimen for this auspicious multimodal approach.

## 4. Materials and Methods

### 4.1. Cell Lines

To determine the effectiveness of several therapies on MPM, three immortalised human MPM cell lines were used. These were kindly provided to us by Prof. Dr. Paul Baas from the Netherlands Cancer Institute (NKI, Amsterdam, The Netherlands). They represent the two most important histological subtypes that are reported in MPM. The NCI-H2818 and NCI-H2795 cell lines represent the epithelioid subtype, which is the most common, while the NCI-H2731 belongs to the most lethal subtype. To keep them in a healthy and controlled environment, cell lines were cultured in flasks containing DMEM/F-12 Glutamax supplemented with 10% fetal bovine serum (FBS) (Thermo Fisher Scientific, Merelbeke, Belgium). Cells were harvested with 0.05% trypsin (Thermo Fisher Scientific, Merelbeke, Belgium).

### 4.2. Peripheral Blood Mononuclear Cells (PBMC)

PBMC were isolated out of blood samples from healthy donors. They consist of lymphocytes (including T cells, B cells and natural killer cells), monocytes, macrophages and dendritic cells (DCs). Isolation of these PBMC was done using Ficoll density gradient centrifugation. For this technique, 25 mL of blood acquired from a healthy donor is carefully dispensed on top of a Ficoll-Paque (GE Healthcare Life Sciences, Diegem, Belgium) in a 50 mL tube. After 20 min of centrifugation at 2100× *rpm* in a swing out rotor (without brake), mononuclear cells form a distinct band at the sample/medium interface. After removing the plasma layer on top of the PBMC, PBMC were transferred into a 50 mL tube and dissolved in phosphate-buffered saline (PBS)/ethylenediaminetetraacetic acid (EDTA). After a second centrifugation step (10 min at 2100× *rpm*), the acquired pellet was resuspended in FBS supplemented with 10% of the cryoprotectant dimethyl sulfoxide (DMSO) for freezing. Subsequently, cryovials contained in a Mr. Frosty^TM^ freezing container filled with isopropanol were placed in a minus 80 °C freezer.After one night at minus 80 °C the vials were stored in liquid nitrogen. Later on, the healthy donor PBMC were used for allogeneic co-cultures with MPM cell lines to determine to treatment schedule and the therapeutic potential of our different combination strategies. Therefore, PBMC were thawed in RPMI-1640 (containing 10% FBS + 10 mM l- glutamine) supplemented with 5 μL DNase (prevents clumping of the DNA from dead cells) and incubated overnight at 37 °C. All the reagents were bought from Thermo Fisher Scientific (Diegem, Belgium).

### 4.3. Sulforhodamine B (SRB) Assay

An SRB-assay was used to determine cell proliferation and to measure treatment-induced cytotoxicity.A cell-specific concentration that provides 80% confluence on the day of analysis was determined for our experiments. With 80% confluence at the analysis, the cells were able to proliferate over time and they are still able to grow, meaning they are still in good condition for analysis. Different concentrations are plated in triplicate in a 96-well plate and incubated overnight at 37 °C. Surrounding empty wells are filled with water to prevent evaporation. The exact quantity of cell suspension needed to achieve the right concentration was determined after cell counting with the Cell Scepter (Millipore). After the cells were grown for 72 h and incubated at 37 °C, the SRB-assay was performed. The assay is based on the capacity of SRB (pink protein dye) to electrostatically – and pH dependently – bind on protein residues of cells fixated to the culture plate after a one-hour incubation in trichloroacetic acid at 4°C [57]. With the use of weak bases, e.g., trisaminomethane, fixated cells can be solubilized for the optic density (OD) measurement at 540 nm with Bio-Rad iMARK^®^ microplate absorbance reader. The OD value will depend on the intensity of the SRB staining which is a measure for the cell density within a well. Based on the OD measurement and microscopically examined cell proliferation, cell concentration ensuring 80% confluence and its corresponding OD value was determined.

After determining the right cell concentration for further experiments, the IC-values of our chemotherapeutics were determined using the SRB-assay. The MPM cells are plated in a 96-well plate and treated for 72 h after an overnight incubation at 37 °C. The chemotherapeutic agents (Selleck Chemicals, Huissen, The Netherlands) investigated in this study are cisplatin (concentration range: 0–50 μM), pemetrexed (concentration range: 0–10 μM) and oxaliplatin (concentration range: 0–50 μM). Experiments were executed in triplicate and repeated three times. Untreated wells and their corresponding OD values were used as control group (representing 100% tumor cell survival) and background staining was excluded from the analysis by subtracting the blanc OD value (measured in a well that only contained cell medium) from the measured values.

After determining the different IC-values IC20, IC40, IC50, IC60 (concentrations providing 20%, 40%, 50% and 60% tumor cell death) with the WinNonlin^®^ software (Pharsight, Mountain View, USA) for each chemotherapy, specific concentrations were selected for further experiments. This selection was based on what has been described in literature and on our obtained results.

### 4.4. Allogeneic Co-cultures

In order to determine the best treatment schedule for combining chemotherapy with immunotherapy, flow cytometry (FCM) was used to examine the effect of the chemotherapeutics on the expression of the immune checkpoints and their ligands. For the experimental set-up, shown in Figure 5, MPM cells were cultured in T75-culture flasks. At 80% confluence, the MPM cells were placed in co-culture with healthy donor PBMC at a 10:1 (Effector: Target) ratio and treated with chemotherapy. PBMC were added to consider the effect that immune cells in the tumor microenvironment might have on the expression of ICPs. After 72 h, PBMC and MPM cells were harvested and transferred into FACS tubes for multicolor FCM on a BD FACSAria II flow cytometer (BD BioSciences, Erembodegem, Belgium).

### 4.5. Flow Cytometry

Our experiments included phycoerythrin (PE)-conjugated Abs against PD-1, PD-L1, PD-L2, TIM-3, LAG-3, galectin-9 and the LIVE/DEAD aqua viability staining. An allophycocyanin (APC)-conjugated Ab against CD45 was also included to distinguish CD45^+^ PBMC from CD45^−^ MPM cells. Corresponding isotype controls for the ICPs were used to take aspecific binding of the Abs into account. Untreated cells were used as a control to determine the baseline expression of the different ICPs. All antibodies were bought by BD Biosciences (Erembodegem, Belgium), except for the LIVE/DEAD aqua (Thermofisher Scientific, Merelbeke, Belgium) and staining was performed according to the manufacturer’s protocol.

To determine the immune checkpoint expression on the PBMC and the MPM cells a specific gating strategy was followed (Figure 6). First, cell debris was excluded based on the scatter profile, followed by exclusion of death cells based on the LIVE/DEAD staining. Within the viable cell population, MPM cells and PBMC were distinguished based on the expression of CD45. Next, Overton percentages and mean fluorescence intensity (MFI) values were extracted from FlowJo^TM^ version 10.4 (TreeStar Inc., Ashland, USA). The overton algorithm calculates the percentages of cells expressing the immune checkpoints (positive cells), while the MFI represents the intensity of the immune checkpoint expression on one cell. For further analysis the delta MFI (ΔMFI) values were used (i.e., the MFI from the isotype control subtracted from the MFI of the stained sample). Comparing overton percentages as well as ΔMFI values between the untreated group and the chemotherapy treated groups provides detailed information on the influence of chemotherapy on immune checkpoint expression.

### 4.6. Statistics

All experiments were performed three times. Results are presented as mean ± standard deviation. Statistical significance was determined by a two-way ANOVA test, followed by a Tukey post-hoc (SPSS software version 23, SPSS Inc., Brussels, Belgium). *p* ≤ 0.05 were considered to be statistically significant.

## 5. Conclusions

Our data show that the expression of immune checkpoints (PD-1, LAG-3 and TIM-3) and their ligands (PD-L1, PD-L2 and galectin-9) on MPM cells as well as PBMC is mostly downregulated or unaltered when treated with different chemotherapeutic agents. If chemotherapy leads to a decreased expression of immune checkpoints it might be better to first administer immune checkpoint blocking Abs. Therefore, an adjuvant setting maynot be the best combination sequence. However, it should be kept in mind that the expression seems to be very dynamic and is most likely patient-dependent. Therefore, more extensive research is required to define a good treatment schedule for chemo-immunotherapy.

Given the immunomodulatory capacities of cisplatin, oxaliplatin and pemetrexed, these are interesting chemotherapeutic agents to combine with immunotherapy. Our data show that all MPM cell lines were most susceptible to cisplatin single agent treatment. Therefore, immune checkpoint blocking Abs will most certainly be interesting to combine with cisplatin, especially when administered in the neoadjuvant setting (first immune checkpoint blocking, then chemotherapy), due to its suspected potential to downregulate the immune checkpoint expression.

## Figures and Tables

**Figure 1 ijms-20-04182-f001:**
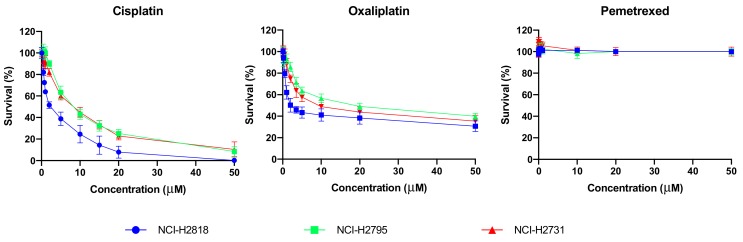
Dose-response curves of cisplatin, oxaliplatin and pemetrexed treatment. Each graph illustrates the survival curve of our three MPM cell lines after 72 h of treatment with either cisplatin, oxaliplatin or pemetrexed. The error bars indicate the standard deviation (*n* = 3). Statistical analysis showed significant differences for cisplatin (*p* = 0.001–0.020) and oxaliplatin (*p* = 0.001–0.009) sensitivity of the different cell lines.

**Figure 2 ijms-20-04182-f002:**
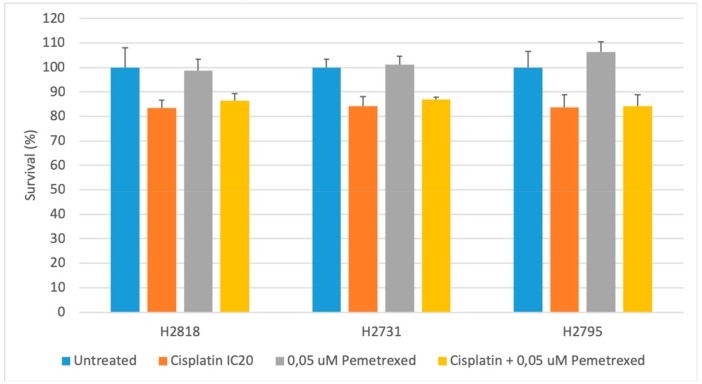
Cytotoxic effect of a combination of cisplatin and pemetrexed. The graph represents the percentage cell survival of three MPM cell lines (NCI-H2818, NCI-H2731 and NCI-H2795) after 72 h of chemotherapeutic treatment with either cisplatin, pemetrexed or cisplatin + pemetrexed compared to an untreated control group. No significant differences were found between the different treatments. Error bars represent the standard deviation (*n* = 3).

**Figure 3 ijms-20-04182-f003:**
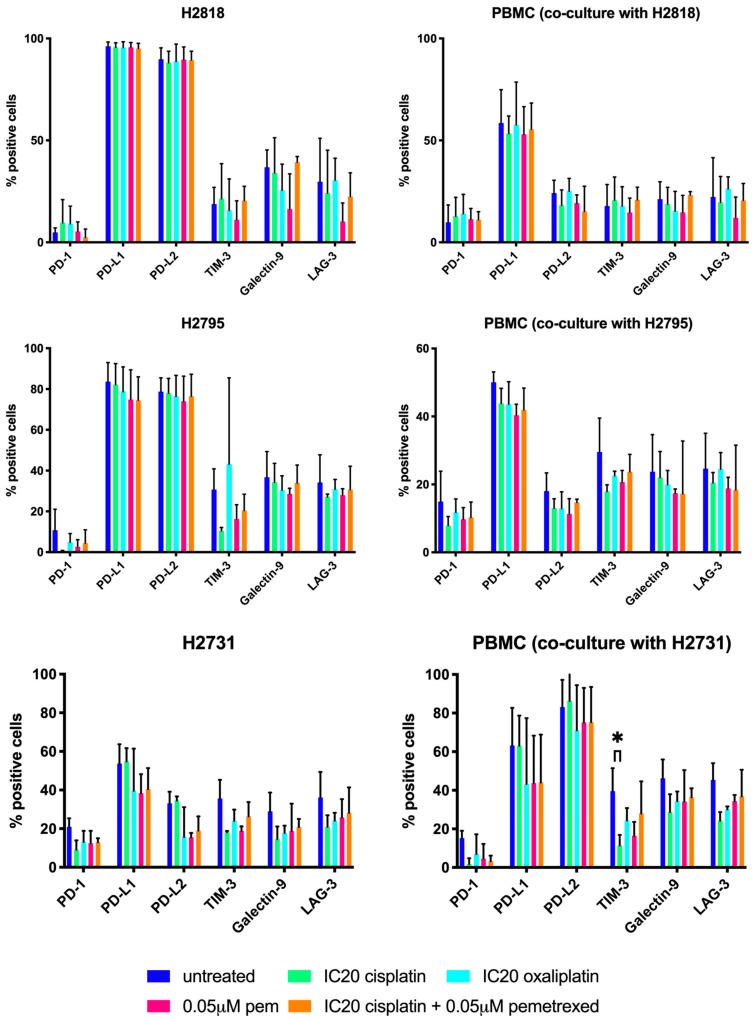
Influence of chemotherapeutics on immune checkpoint expression on MPM cell lines and PBMC in co-culture (overton percentages). Bar charts of mean overton percentages representing the percentages of NCI-H2818, NCI-H2795, NCI-H2731 and corresponding PBMC that express the immune checkpoints or ligands. Following chemotherapy. Error bars represent the standard deviation (*n* = 3). * *p* < 0.05: significant difference in % of cells expressing immune checkpoints or ligands * *p* < 0.05: significant difference in immune checkpoint expression. Isotype controls were used to consider aspecific binding of the flow cytometry staining.

**Figure 4 ijms-20-04182-f004:**
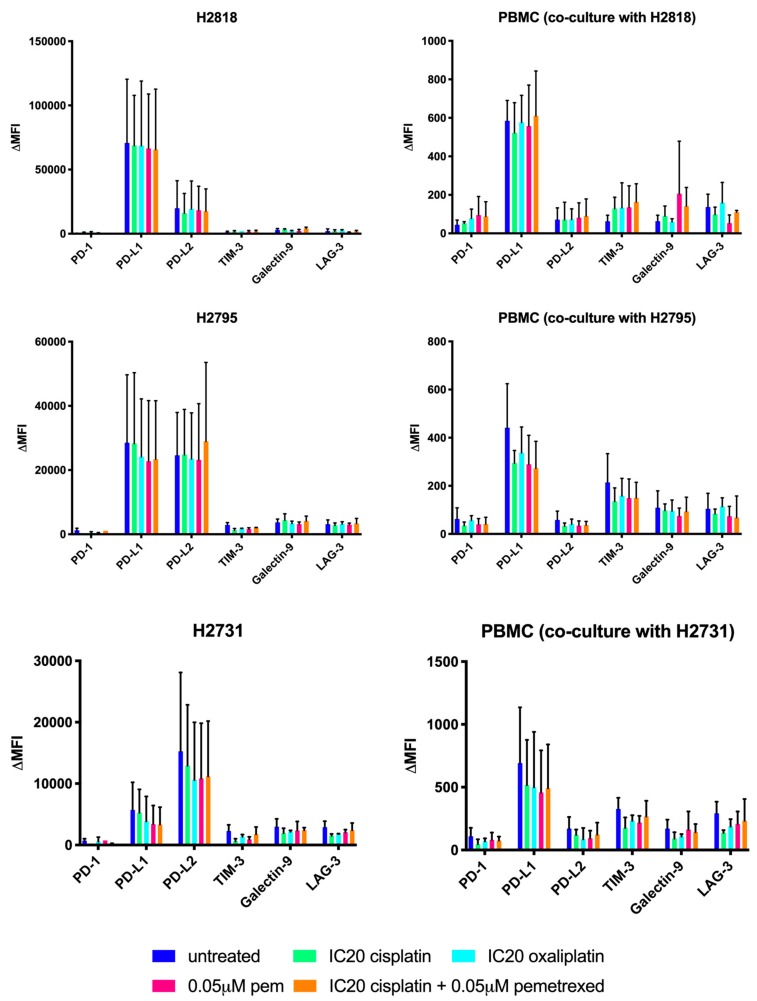
Influence of chemotherapeutics on immune checkpoint expression on MPM cell lines and PBMC in co-culture (ΔMFI values). Bar charts of mean ΔMFI values representing the expression of the immune checkpoints or ligands on NCI-H2818, NCI-H2795, NCI-H2731 and corresponding PBMC. Expression is determined after treatment. Error bars represent the standard deviation (*n* = 3). *p* < 0.05: significant difference in immune checkpoint or ligand expression. Isotype controls were used to consider aspecific binding of the flow cytometry staining.

**Figure 5 ijms-20-04182-f005:**
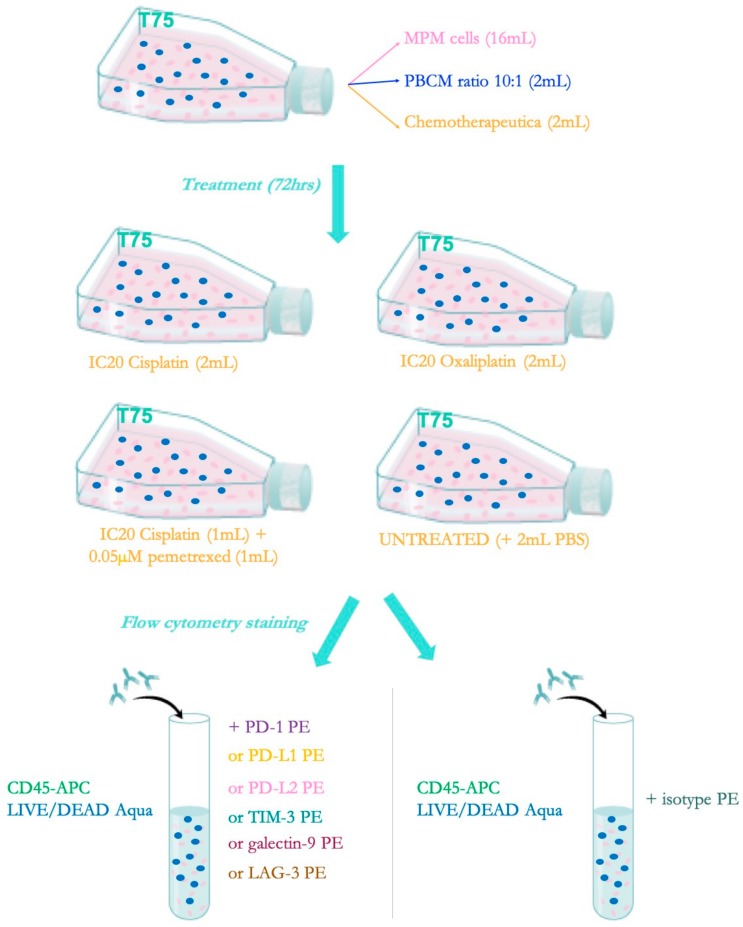
Experimental set-up allogeneic co-culture assay with flow cytometric read-out for chemotherapy-induced modulation of ICP expression. Co-cultures of MPM cells and PBMC were either untreated or treated with cisplatin, pemetrexed, cisplatin + pemetrexed or oxaliplatin to determine the influence of the treatment on immune checkpoint expression. PBMC and MPM cells were harvested and transferred into tubes. For each condition, the PBMC and MPM cells were stained with PE-labelled antibodies against the targets of interest: PD-1, PD-L1, PD-L2, TIM-3, LAG-3, and galectin-9. CD45 and LIVE/DEAD Aqua was added to each well to distinguish MPM cells from PBMC and viable from dead cells, respectively.

**Figure 6 ijms-20-04182-f006:**
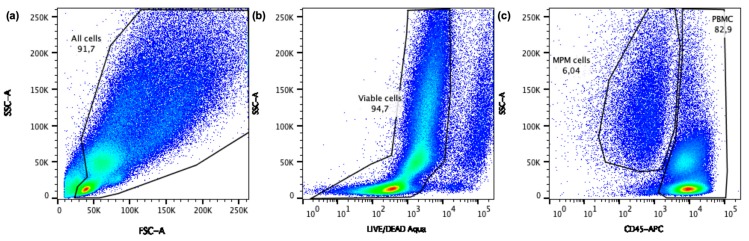
Flow cytometry gating strategy. The dot plots show the subsequent gates that were drawn to analyze the immune checkpoint expression on MPM cells as well as on healthy donor PBMC. (**a**) Debris was excluded based on SSC versus FSC, (**b**) then dead cells were eliminated based on SSC versus Alexa Fluor 430, (**c**) and finally MPM cells and PBMC can be distinguished within the viable population based on the expression of CD45.

**Table 1 ijms-20-04182-t001:** Inhibitory concentrations of cisplatin and oxaliplatin resulting in 50% survival.

Chemotherapy	Cell Line	IC50	*p*-Values		
			NCI-H2818	NCI-H2795	NCI-H2731
Cisplatin	NCI-H2818	2.31 ± 0.34		0.008	0.007
	NCI-H2795	7.78 ± 0.44	0.008		1.000
	NCI-H2731	7.89 ± 0.44	0.007	1.000	
Oxaliplatin	NCI-H2818	3.47 ± 1.52		0.001	0.030
	NCI-H2795	18.23 ± 3.98	0.001		0.012
	NCI-H2731	9.77 ± 2.33	0.030	0.012	

Overview of the IC50 values for each cell line. IC50 values were calculated by Winonlin software Standard deviations of the IC50 values are given for each cell line.

**Table 2 ijms-20-04182-t002:** Inhibitory concentrations of cisplatin and oxaliplatin used for subsequent experiments.

Cell Line	IC20 (µM)		
	*Cisplatin*	*Oxaliplatin*	*Pemetrexed* ^[18]^
NCI-H2818	0.50	0.17	0.05
NCI-H2795	2.70	1.86	0.05
NCI-H2731	2.34	1.01	0.05

Overview of the IC20 values that are used for our experiments. IC20 values were calculated based on the IC50 values.

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
