# Peer review of "Building a Bridge between Chemotherapy and Immunotherapy in Malignant Pleural Mesothelioma: Investigating the Effect of Chemotherapy on Immune Checkpoint Expression"

_ijms, 2019, doi:10.3390/ijms20174182_

Round 1
Reviewer 1 Report
Mesothelioma is an aggressive and increasing cancer in industrialized countries.
The prognosis remain poor despite the multimodal treatments.
The authors, using three different MPM cell lines representative for epithelioid and sarcomatoid subtypes, studied the effect of first-line chemotherapy and the immunogenic chemotherapeutic agent oxaliplatin on the expression of the immune checkpoint.
The results showed that cisplatin, oxaliplatin and pemetrexed are interesting chemotherapeutic agents to combine with immunotherapy, given their immunomodulatory capacity. However the expression would seem to depend on the patient or of the morphologic aspects of the mesothelioma.
- I would therefore suggest that the authors include in discussion and so in the references:
- in recent works Kadota et al (2018) and Pelosi et al (2018), observed that survival in patients with pleural mesothelioma despite chemotherapeutic treatments may be conditioned not only by a histological subtype but also a standardized histological grading system.
Author Response
authors, using three different MPM cell lines representative for epithelioid and sarcomatoid
subtypes, studied the effect of first-line chemotherapy and the immunogenic chemotherapeutic
agent oxaliplatin on the expression of the immune checkpoint.
The results showed that cisplatin, oxaliplatin and pemetrexed are interesting chemotherapeutic
agents to combine with immunotherapy, given their immunomodulatory capacity. However the
expression would seem to depend on the patient or of the morphologic aspects of the
mesothelioma.
- I would therefore suggest that the authors include in discussion and so in the references:
- in recent works Kadota et al (2018) and Pelosi et al (2018), observed that survival in patients
with pleural mesothelioma despite chemotherapeutic treatments may be conditioned not only
by ahistological subtype but also a standardized histological grading system.
Dear Reviewer,
Thank you for reviewing our manuscript. We appreciate your valuable remark. We do agree
that the expression of immune checkpoints is patient dependent and varies among the different
histological subtypes, as shown by our previously published results (Marcq et al.,
Oncoimmunology 2017; Marcq et al., Oncotarget 2017).
Our current data show that response to chemotherapy treatment might not only depend on the
histological subtype. The references that you recommended report that not only the subtype
but also nuclear grading of the tumor conditions survival. It might be possible that this also
influences response to chemotherapy and therefore we added this in our discussion (page 11,
lane 1-2). The references are also added to our list of references (references 32 and 33).
Changes in the manuscript text are highlighted using the “Track Changes” function.
Reviewer 2 Report
In this study, the authors want to investigate the effects of the chemotherapeutic agents used to treat patients with MPM, on the expression of immune check point markers. The results will help to evaluate whether an immunotherapeutoc treatment combined to chemotherapy could have a better effect on mesothelioma developement.
The scientific question is interesting and clearly presented in the introduction.
But the methods and results obtained are very insufficient to be published. It lacks a lot of control. For example, surprisingly, the three cell lines are not sensitive to the chemotherapeutic agent pemetrexed. The results didn't show a positive control of the potency of the pemetrexed using a sensitive cell line studied in the same condition.
The treatment of co-cultures was only performed for 24H. It seems to be very short to induce change in the expression of ICP markers. There is no control to evaluate the efficiency of the flow cytometry to check changes in the expression of ICP markers.
The method part is clearly insufficiently described. It's impossible to reproduce experiments.
In summary, these results are very preliminary and can not be published at this stage.
Author Response
Dear Reviewer,
Thank you for your valuable remarks on our manuscript. We have revised our manuscript
based on your recommendations. Listed below is our point-by-point response to your
comments. Changes in the manuscript text are highlighted using the “Track Changes” function
as suggested by the editors of the journal.
§ The methods and results obtained are very insufficient to be published. It lacks a lot of
control. For example, surprisingly, the three cell lines are not sensitive to the
chemotherapeutic agent pemetrexed. The results didn't show a positive control of the
potency of the pemetrexed using a sensitive cell line studied in the same condition.
To our knowledge, so far nothing has been reported in literature on in vitro treatment of
mesothelioma cell lines with pemetrexed. The only data that can be found are from clinical
studies reporting improved survival in mesothelioma patients treated with the combination
of cisplatin + pemetrexed compared to cisplatin monotherapy. Therefore, we could not
include a pemetrexed sensitive mesothelioma cell line as a control.
We decided to test the mesothelioma cell lines that were available in our lab and indeed
none of these showed any response to pemetrexed treatment. In order to exclude any
coincidence, we tested a new lot of pemetrexed and we even optimized the treatment
conditions as recommended by the company. Because activity of the salvage pathway of
the cells can make it impossible to obtain IC50 values under routine conditions we were
recommended to use dialyzed sera. However, still none of our cell lines showed any
response to pemetrexed thereby supporting our previous observations.
To investigate the IC50 value we based ourselves on data from our colleagues who
reported pemetrexed response in a NSCLC cell line A549 (Wouters et al., BMC Cancer
2010). Because no response was observed using similar concentration we even increased
our doses more than 10 times (>2μM pemetrexed). However, still no response was seen,
supporting the fact that our mesothelioma cell lines just not respond to pemetrexed.
Taken together, since the goal of our research was to assess the effect of clinically relevant
doses of different chemotherapeutics on the expression of ICPs in mesothelioma, we did
not increase the concentration of pemetrexed nor did we test it in other cancer types.
§ The treatment of co-cultures was only performed for 24H. It seems to be very short to
induce change in the expression of ICP markers. There is no control to evaluate the
efficiency of the flow cytometry to check changes in the expression of ICP markers.
We thank the reviewer for pointing this out as the time points were mistakenly written in
our first manuscript. Tumor cells were plated and 24 hours later, they were put in co-culture
with PBMC and treated with chemotherapy for 72 hours. We have corrected this error
throughout the manuscript and adjusted it in figure 5 on page 16.
The efficiency of flow cytometry to detect changes in ICP expression has been previously
confirmed in our lab. As illustrated for PD-L1 (CD274) in the figure below, we saw an
upregulation of PD-L1 after 24 hours incubation with interferon-gamma (IFNg). These data
confirm that flow cytometry can be used to evaluate the expression of ICPs.
Influence of IFNg on the expression of PD-L1. MPM cells were incubated for 24 hours with
IFNg after which the expression of IFNg was investigated using flow cytometry.
§ The method part is clearly insufficiently described. It's impossible to reproduce
experiments.
We went through the methods again and added information about the reagent’s suppliers
(page 15, lane 38-39) so that the readers can reproduce the experiments more easily.
Furthermore, we believe that all the essential information to reproduce our data is included.
In brief:
1. we mentioned the cell lines that are used, their origin and cell culturing conditions
(page 14, lane 9-15)
o we describe our protocol for the isolation of PBMC and the different reagents and
centrifugation steps (page 14, lane 18-33)
2. we describe extensively the assay and software that we used to determine the IC50
values of our chemotherapeutic agents (page 14, lane 36-48; page 15, lane 1-19)
3. the set-up of our co-culture is described (page 15, lane 22-30) and clarified in figure
5 (page 17)
4. we describe our flow cytometry staining protocol (page 15, lane 33-39) and our
gating strategy (page 17, lane 11-18; page 18, lane 1-3) which is clarified in figure
6 (page 18)
5. we describe our statistical analysis and mention which statistically significant pvalue
that we used (page 17, lane 13-18)
We hope these changes and presented information would suffice. We are also open to
more specific requests to improve the readability and reproducibility of the experiments
performed here.
§ In summary, these results are very preliminary and cannot be published at this stage.
Although our results might seem to be preliminary we strongly believe that they are worth
publishing. With the promising results of immune checkpoint blocking therapies in several
cancer types and the rising interest in combination strategies, the next logical step would
be to look into combinations with the 1st line treatment. In the case of mesothelioma this is
chemotherapy (cisplatin + pemetrexed). However, one cannot design a treatment schedule
without any rationale behind it. Therefore, we decided to look at the effect of chemotherapy
on the expression of our immunotherapeutic targets. Despite the fact that our experiments
did not lead to conclusive results regarding a rational treatment design, we believe that our
manuscript will be of interest for those who are investigating combination strategies with
immune checkpoint blockade and who want to take the effect of the tumor
microenvironment into account (i.e. influence of immune cells).
In summary, the design of our experiment presented here follows the logical progression
of preliminary (screening) tests for combination therapies which other research groups may
also investigate: To advance progress in this area, publication of our study can limit
repetition of similar experiments and stimulate new ideas.
Reviewer 3 Report
The present manuscript contains many repetitions in introduction and discussion, whereas there is not an explanatory paragraph in introduction section concerning the expression of immune checkpoints in the different types of resistant cancer. Moreover, some sentences regarding the common knowledge of the chemotherapy/radiotherapy are unnecessary (e.g. lanes 11-19, 21-27, 47-50 page 2), whereas others should be improved (e.g. . lanes 42-43, page 1; 51-52 page 2). The aim is not well described, as I suppose that the authors wanted investigate whether their treatments were effective to downregulate ICPs expression, thus miming the ICPs blocking to increase drugs treatment efficacyand overcome MDR.The design of the study contains some gaps. The results are not well explained and there are so many imperfections that make the manuscript weak in its content to support final conclusion that, moreover, is not well written in conclusions paragraph (lanes 21-24 page 22). The title is too vague and it seems appropriate for a review, the discussion is not pertinent on the basis of the obtained data and, obviously, the study lacks of a mechanism of action.
Particularly, my attention is focused on the following concerns and doubts.
It is too general the rational to which the authors refer to choose the doses of chemotherapeutic drugs “Based on data derived from literature” (lane 10 page 3). Indeed, it is well known that each cancer cell line has its time and dose response to drugs which should be tested at the beginning of the study as the authors have done.
In Figure 1 the unit of concentrations are illegible.
What is the meaning for “was more susceptible” (lane 15 page 2) or for "a trebd towards.." (lane20 page 21)? It is statistically significant or not? Moreover, the presented data in figures and table 1 are totally lack of statistical analysis and of its representation on each panel. It is very difficult understand what the authors wanted to show.
What is the meaning for “We worked with the IC20 values of cisplatin and pemetrexed because we want to use for each compound a dose as low as possible with already an effect for the combination strategy” (lane 6-7 page 4 ), if the authors did not see any statistically cytotoxic effect using combined drugs? What about the cytotoxic effect of oxaliplatin alone? and why the authors did not calculate IC for combined treatments? Also, sentences in lanes 8-12 page 4, in lanes 28-34 page 4 are completely incomprehensible in the data findings.
What is the meaning for “our MPM cell lines were resistant to pemetrexed" (lane 26-27page 3 )? Have the authors any data to support the multidrug resistance phenotype? What happens increasing the dose of pemetrexed? The discussion is not appropriate in pemetrexed multidrug resistance mechanisms. It is very easy and inexpensive detect the accumulation of drugs (doxorubicin) to assess whether the cell are really resistant.
Figure 3 (lane 15 page 4 ) should be named as Figure 2.
Figure 5 and 6 are superfluous in their meaning and figure 5 contains some mistakes. What about the reducing time (72 to 24 hours) in treatments?
Finally, I firmly believe that the way in which the experiments were performed (dose, timing, choose of drugs and incubation treatments) was the detrimental point not to obtain some consistent results and thus reach a solid conclusion.
Author Response
Dear Reviewer,
Thank you for your valuable remarks on our manuscript. Your profound evaluation has helped
us to revise it as described below by our point-by-point response to your comments. Changes
in the manuscript text are highlighted using the “Track Changes” function.
The present manuscript contains many repetitions in introduction and discussion, whereas
there is not an explanatory paragraph in introduction section concerning the expression of
immune checkpoints in the different types of resistant cancer. Moreover, some sentences
regarding the common knowledge of the chemotherapy/radiotherapy are unnecessary (e.g.
lanes 11-19, 21-27, 47-50 page 2), whereas others should be improved (e.g. . lanes 42-43,
page 1; 51-52 page 2). The aim is not well described, as I suppose that the authors wanted
investigate whether their treatments were effective to downregulate ICPs expression, thus
miming the ICPs blocking to increase drugs treatment efficacy and overcome MDR.The design
of the study contains some gaps. The results are not well explained and there are so many
imperfections that make the manuscript weak in its content to support final conclusion that,
moreover, is not well written in conclusions paragraph (lanes 21-24 page 22). The title is too
vague and it seems appropriate for a review, the discussion is not pertinent on the basis of the
obtained data and, obviously, the study lacks of a mechanism of action.
We thank the reviewer for the comments here, particularly on the point of multidrug resistance.
We had no intension of addressing MDR in this study and were not aware that this inference
was taken from our manuscript. We have since made changes to the introduction to further
highlight the aim and goal of the study: “In this study, we aimed to investigate the effect of
chemotherapeutic agents on the ICP expression of immune cells and MPM tumor cells in order
to develop a rational treatment schedule for the combination of chemotherapy with ICP
blockade (ICPB).” (page 2, lane 32-35)
Introductory sentences were added to each part of the result sections in order to capture the
goal of each experiment (page 4, lane 25-26; page 7, lane 2-5). Furthermore, we have added
additional information and edited sentence structure to improve readability and understanding.
We believe in the integrity of the results that we obtained, since all our experiments were
repeated 3 times and showed similar results every time.
We have altered the discussion to make more clear links to our obtained data and the
conclusions have also been adjusted. We did not look into mechanisms of action as this was
beyond the scope of our research. By clarifying our study aims and making better connections
between our results with existing literature, we hope the scope and the value of the work
becomes more evident.
The final manuscript was checked by a native English-speaking colleague.
Particularly, my attention is focused on the following concerns and doubts.
§ It is too general the rational to which the authors refer to choose the doses of
chemotherapeutic drugs “Based on data derived from literature” (lane 10 page 3). Indeed,
it is well known that each cancer cell line has its time and dose response to drugs which
should be tested at the beginning of the study as the authors have done.
We have added additional information, including concentration ranges, to provide more
details on our dose decisions; page 4 (lane 27-30).
§ In Figure 1 the unit of concentrations are illegible.
We enlarged the font size of the “μ” symbol as shown on the figure below.
The adjusted figure is added in the manuscript on page 4.
§ What is the meaning for “was more susceptible” (lane 15 page 2) or for "a trend towards.."
(lane20 page 21)? It is statistically significant or not? Moreover, the presented data in
figures and table 1 are totally lack of statistical analysis and of its representation on each
panel. It is very difficult understand what the authors wanted to show.
Thank you for your comment. We have included more detailed information about the
statistical analysis used in this study on page 17 (lane 13-18). The key message of this
paragraph in the manuscript is that our MPM cell lines displayed different sensitivity /
susceptibility to chemotherapy and therefore their IC values differ. We have tried to clarify
the message in the manuscript (page 2, lane 32-39).
Statistical analysis was performed on data in figure 1, but asterisks to display statistical
significance were excluded to improve cluttering in the graph. We have added the range
of p-values from the statistical analysis in the text (page 4, lane 34-36) and in the figure
legend.
Data from the original table 1 are now split into two new tables on page 5 and 6. Table 1
on page 5 shows the IC50 values and their corresponding p-values that indicate significant
differences between our cell lines. The p-values are also mentioned in the text (page 5,
lane 8-11). Table 2 on page 6 shows the IC20 values that were selected for our further
experiments.
§ What is the meaning for “We worked with the IC20 values of cisplatin and pemetrexed
because we want to use for each compound a dose as low as possible with already an
effect for the combination strategy” (lane 6-7 page 4 ), if the authors did not see any
statistically cytotoxic effect using combined drugs? What about the cytotoxic effect of
oxaliplatin alone? and why the authors did not calculate IC for combined treatments? Also,
sentences in lanes 8-12 page 4, in lanes 28-34 page 4 are completely incomprehensible
in the data findings.
Thank you for your comment and we have clarified our rationale on page 5 (lane 25-29).
Since the goal was to find a synergistic combination strategy of chemotherapy and
immunotherapy, we selected low doses of chemotherapy to better identify the effects of
combination with immune checkpoint blockade. However, due to the unexpected effects of
chemotherapy on immune checkpoint expression we observed, we did not pursue further
cytotoxicity experiments with the combination strategy. Since cisplatin and pemetrexed are
used as first-line treatments, the treatment of cisplatin + pemetrexed was viewed as a
single treatment and IC values for the chemotherapeutic combination were not calculated.
Our graph in figure 1, represents the sensitivity of all our cell lines to oxaliplatin
monotherapy.
We clarified the sentences on:
Page 4, lane 8-12: we have rewritten the sentences here to help improve the
reader’s understanding (page 6, lane 8-14)
Page 4, lane 28-34: since some of the reported observations were not essential for
our final conclusion we removed them out of the text. Description of figures 3 and
4 was shortened in order to simplify our message (page 7, lane 14-33)
§ What is the meaning for “our MPM cell lines were resistant to pemetrexed" (lane 26-27page
3 )? Have the authors any data to support the multidrug resistance phenotype? What
happens increasing the dose of pemetrexed? The discussion is not appropriate in
pemetrexed multidrug resistance mechanisms. It is very easy and inexpensive detect the
accumulation of drugs (doxorubicin) to assess whether the cell are really resistant.
With this sentence we mean that our cell lines were not sensitive to pemetrexed, which is
also shown by the flat survival curves on figure 2C (adjusted on page 5, lane 12). The
confusion may in part be due to the confusion in the introduction and discussion. By
addressing these points in the above sections, we hope the scope and aim of our study
becomes clear.
We have no data regarding the multidrug resistance phenotype of our cell because this is
beyond the scope of our research question. Regarding increasing the dose of pemetrexed,
investigated a broad range of concentrations and even with our highest concentration -
which is 10 times higher than concentrations used for pemetrexed sensitive cell lines
(Wouters et al, BMC Cancers 2010) - we did not see any effect on cell survival. Since our
focus was to obtain results with a translational value for new combination strategy in the
clinic, increasing the dose beyond this point becomes irrelevant, as patients are treated
with a lower dose of pemetrexed in comparison to cisplatin.
§ Figure 3 (lane 15 page 4 ) should be named as Figure 2.
We have adjusted the numbering.
§ Figure 5 and 6 are superfluous in their meaning and figure 5 contains some mistakes. What
about the reducing time (72 to 24 hours) in treatments?
In light of the comments from other reviewers regarding the details of our materials and
methods, we have decided to keep figure 5 and 6. We also believe in the added value of
visualizing our experimental set-up and gating strategy as it might be of ease for people
who want to reproduce our experiments. We hope the reviewer understands our reasoning,
and if needed, these figures can be added into supplementary material.
We apologize for mixing up time point throughout the manuscript: 24 hours after the cells
were plated out they were put in co-culture with PBMC and treated with chemotherapy for
72 hours. We have corrected this in the manuscript and adjusted figure 5 on page 16.
§ Finally, I firmly believe that the way in which the experiments were performed (dose,
timing, choose of drugs and incubation treatments) was the detrimental point not to
obtain some consistent results and thus reach a solid conclusion.
Multiple varying results have been reported by other groups, and the discrepancies in our
results have been addressed in our discussion (page12, lane 12-22). We believe in the
integrity of our work, since all our experiments showed similar results after 3 individual
repetitions. The design of our experiment also was carefully rationalized:
• Doses were chosen based on reports in literature, and we made sure our working
concentrations for the chemotherapeutics were within the plasma concentration
range of the maximum tolerated dose reported in patients in Phase I trials. This
was to further ensure our results would be more clinically relevant.
• The timings and incubation times were chosen to ensure we see the effect of
treatment as earlier timepoints may not be sufficient. We recognized that the
confusion here may be due to our mistake in the time points which we have now
corrected (addressed above).
• The drugs we chose were most clinically relevant as cisplatin + pemetrexed are
the first line treatment used in the clinic for MPM patients. Oxaliplatin was chosen
for its capacity to induce immunogenic cell death and also because of its close
chemical similarity to cisplatin.
We are aware of the fact that no solid conclusion can be drawn from our results but we
believe in the added value of also publishing negative results. These may contribute to new
insights and optimization of experiment for people planning to investigate similar things in
the future.
Round 2
Reviewer 2 Report
The methods and results obtained are still very insufficient to be published. The only data showed are flow cytometry results with a lot of variability in experiments. Due to this lack of reproducibility, it's impossible to conclude anything about the expression of ICP on MPM cells after chemotherapeutic treatment.
To be published the results have to be completed at least with:
MPM cells sensitive to pemetrexed such as MSTO-211H easily available from ATCC. In contrast to your response, several articles have shown the sensitivity of mesothelioma cell lines to pemetrexed. For example, Cortes-Dericks L et al., 2010; Sato Y et al., 2018; Moriya H et al., 2013. study of the expression of ICP on MPM cells (+/- PBMC) using other methods than flow cytometry such as western blot and RT-qPCR. Expression of cell surface markers may be some times difficult to detect by flow cytometry due to difficulty in the method and/or high turn-over of the protein on the cell surface. So using other methods may help to detect a change.Maybe chemotherapeutic treatments don't change the expression of ICP. It's also possible and recommended to publish "negative" results. But it's more difficult to publish "negative" results because you have to prove that these results are not due to a problem in method but really due to the absence of changes in your sample. So the best way is to find a positive control showing changes in ICP expression in the same experimental conditions.
The method part stays insufficiently described. Some standards are used to write the Methods part. For example, you have to indicate the company for each reagent (SRB-assay, Ficoll-paqe, ...), to use international unit (for example, centrifugation rate should be stated in "g" and not in "rpm" that is dependant of the centrifuge used), to indicate the number of cells (MPM cells, PBMC, ...) cultured in which plate (T75, 96-wells plate, ...), the number of events used to perform your flow cytometry analysis, the reference of the antibodies used to label the cells, the concentration, the time of incubation, ...
In summary, you have first to perform additional experiments to present robust and clear results available for publication, and second to work on the article.
Reviewer 3 Report
In my opinion the paper has been improved.
However the title is still not appropriated to the results.